# A Comprehensive Genomic Analysis of Nucleophosmin (NPM1) in Acute Myeloid Leukemia

**DOI:** 10.3390/cancers17162710

**Published:** 2025-08-20

**Authors:** Osama Batayneh, Mahmoudreza Moein, Alexandra Goodman, Devashish Desai, Dean Pavlick, Chelsea Marcus, Caleb Ho, Russell Madison, Richard S. P. Huang, Jeffrey S. Ross, Teresa Gentile, Zheng Zhou, Krishna Bilas Ghimire

**Affiliations:** 1Department of Medicine, Division of Hematology/Oncology, SUNY Upstate Medical University, 750 East Adams Street, Syracuse, NY 13210, USA; 2SUNY Upstate Medical University, Syracuse, NY 13210, USA; 3Foundation Medicine, Cambridge, MA 02141, USA

**Keywords:** acute myeloid leukemia, nucleophosmin, genomic alterations, genomic study, outcomes

## Abstract

Mutations linked to therapy targets in AML, such as *FLT3* and *IDH1/2,* are more frequently identified in NPM1mut AML, whereas *KMT2A*, *TP53,* and myelodysplastic-related mutations are more frequently identified in NPM1wt AML. *DNTM3A* and *PTPN11,* which correlate with inferior outcomes, are more commonly observed in NPM1mut AML. This genomic landscape study highlights significant genomic differences between NPM1mut and NPM1wt AML patients, which may enrich our understanding of the molecular profile and mutation clusters in AML.

## 1. Introduction

Acute Myeloid Leukemia (AML) is an aggressive, heterogeneous hematologic malignancy with diverse genetic abnormalities that accounts for approximately 80% of acute leukemia cases and is the most common acute leukemia in adults [1,2]. Leukemia develops from the serial acquisition of somatic mutations in hematopoietic stem and progenitor cells with the capacity to regenerate the neoplastic clone [3,4].

AML can occur at any age; however, it predominantly affects older adults, with a median age at diagnosis of 68 years [5], and accounts for 1.1% of new cancer diagnoses in the United States [6]. Recent advances and our better understanding of the pathogenesis, molecular testing, and the development of novel therapies have brought us to a new era regarding the diagnosis, classification, and treatment of patients with AML [7].

AML is defined by a range of recurrent genomic mutations that influence disease phenotype, therapeutic response, relapse risk, and survival. Within the past few decades, genomic studies based on next-generation sequencing (NGS) have further dissected the molecular profile of AML and changed the landscape of AML treatment [3,4]. An international expert panel from the European LeukemiaNet (ELN) published a well-validated risk stratification tool, which is largely based on comprehensive molecular and cytogenetic analysis at the time of diagnosis for patients receiving intensive chemotherapy [8], as well as an updated molecular risk stratification for patients receiving less intensive chemotherapy [9].

The overall mutational burden in AML averages approximately five recurrent mutations per genome—driver mutations are detectable in 96% of patients with de novo AML, with 86% harboring two or more. Among the most frequently observed mutations in newly diagnosed adult AML are *NPM1* (≈30%), *DNMT3A* (≈20%), and *FLT3* (≈20–25%) [3,4,10,11]. A comprehensive molecular and cytogenetic analysis at diagnosis can impact treatment decisions and intensity [8,9,12,13,14,15], even for older patients [16,17]. This is observed especially in the era of newer and targeted therapies, like liposomal daunorubicin and cytarabine (Vyxeos) for high-risk/secondary AML [18], venetoclax combined with hypomethylating agents [2,19,20,21,22], gemtuzumab ozogamicin (GO) for favorable risk AML [23], *FLT3* inhibitors with or without chemotherapy for AML with *FLT3* mutation [24,25,26], ivosidenib and azacitidine in *IDH1*-mutated acute myeloid leukemia [8,27], *IDH2* inhibition in *IDH*-mutated AML [28], and menin inhibitors targeting driver mutations in AML such as *KMT2A* and *NPM1* [29,30,31,32].

Nucleophosmin (*NPM1*), also referred to as B23, No38, or numatrin, is a highly multifunctional protein encoded by the *NPM1* gene. *NPM1* is identified as a phosphoprotein localized predominantly in the granular component of the nucleolus and is also known for its ability to transfer between the nucleus and cytoplasm [33,34,35]. Point mutations in the *NPM1* gene lead to the aberrant cytoplasmic localization of *NPM1*-associated nuclear proteins into the cytoplasm, including several transcription factors. This leads to suppression of multiple terminal differentiation genes. The suppression of differentiation by *NPM1* enables a leukemic transcription program that is highly dependent on the up-regulated expression of HOXA and MEIS1 genes. Expression of these genes further blocks differentiation and induces long-term proliferation, leading to the leukemic phenotype [36,37]. HOX/MEIS signatures of *NPM1* AMLs overlap with those of *KMT2Ar* leukemias. Maintenance of this transcription signature in *NPM1* cells depends directly on the menin–KMT2A interaction, supporting the role of menin inhibitors in the treatment of *NPM1*-mutated AML [32,38]. Little is known regarding how *NPM1* cells maintain aberrant gene expression.

The aim of this study is to conduct a comprehensive landscape genomic analysis of key genomic alterations in AML and to compare NPM1mut and NPM1wt (“wild type” defined as cases without *NPM1* gene mutations). This comparison seeks to enhance our understanding of AML molecular profile and highlight significant differences in NPM1mut and NPM1wt, which may aid in delineating this mechanism and improving our understanding of mutations in AML.

## 2. Method and Materials

### 2.1. Study Design

Approval for this study, including a waiver of informed consent and a HIPAA waiver of authorization, was obtained from the Western Institutional Review Board (Protocol No. 20152817). A retrospective analysis of peripheral blood specimens was performed on 4206 AML patients from 2019 to 2024 who underwent comprehensive genomic profiling, using the FoundationOne Heme combined hybrid capture-based DNA and RNA sequencing assay. All classes of relevant GA were evaluated, including base substitutions, short insertions and deletions, copy number changes and rearrangements, and fusions. The DNAseq component detects the entire coding region of 405 genes and selects intronic regions in 31 genes known to be clinically and biologically relevant in cancer. The RNAseq component is focused on 265 genes recurrently rearranged in cancer. This assay is validated to a high accuracy, achieved by high, uniform coverage: average median exon depth of 500× (DNA), average on-target distinct pairs ~3 M (RNA). Patient age, biological sex, and genomic ancestry were extracted from accompanying pathology reports. As self-reported race was not available, predominant patient ancestry was determined for each specimen using a custom SNP-based classifier, as previously described [39]. The tumor mutational burden (TMB) is defined as the total number of mutations found in the DNA of cancer cells and is reported as the number of mutations seen in a section of DNA and reported as mutations per megabase (mut/Mb) (a TMB of 10 mut/Mb or greater was referred to as TMB-high) [40]. Homologous recombination deficiency signature (HRDsig) is defined as the inability of a cell to effectively repair DNA double-strand breaks using the homologous recombination repair (HRR) pathway [41], and Microsatellite Stability (MSS) status was determined on at least 1500 loci [42]. The Catalogue of Somatic Mutations in Cancer (COSMIC) was used in order to reflect the underlying mechanisms of mutational processes in each case [43].

Patients who were at least 18 years old, diagnosed with AML, and underwent next-generation NGS were included. The FoundationOne Heme assay was utilized for comprehensive genomic profiling as previously described [44]. Figure 1 demonstrates the genomic sequencing process illustration as per FoundationOne testing.

Based on the genomic profiling results, study participants were categorized into two cohorts: those diagnosed with *NPM1*-mutated (NPM1mut) AML and those with NPM1 wild-type (NPM1wt) AML.

### 2.2. Outcome Definitions

The primary objective of the study is to perform a comprehensive analysis of key genomic alterations in AML and identify genetic differences between the NPM1mut and NPM1wt AML.

### 2.3. Statistical Analyses

The median age of patients was calculated. The *t*-test was performed for univariate analysis of continuous variables, and the chi-square test and Fisher’s exact test were performed for univariate analysis of categorical variables. Categorical data were summarized as proportions and percentages, and continuous data were summarized as means and standard deviations (±SDs). Fisher’s exact test with the Benjamini–Hochberg correction was used to control and reduce the likelihood of false discovery. The Benjamini–Hochberg procedure ranked individual *p*-values from multiple comparisons, then applied an adjusted significance threshold that becomes progressively stricter as the number of tests increases. A *p*-value < 0.05 was considered significant.

## 3. Results

### 3.1. Clinical Characteristics

A total of 4206 cases of AML specimens were included in our final study cohort. A total of 3573 (84.9%) AML cases were NPM1wt, and 633 (15.1%) AML cases were NPM1mut. Female gender was identified at a significantly higher frequency in the NPM1mut cohort compared to the NPM1wt cohort (53.4% vs. 41.5%; *p* < 0.0001). Also, patients in the NPM1mut cohort were slightly older compared to the NPM1wt AML patients (62 years old vs. 60 years old, *p* < 0.0001). More than half of the patients (>60%) were of European descent. *NPM1* genomic alterations were more prevalent among patients of European ancestry (77.1% vs. 68.5%; *p* < 0.0001) and slightly less common among those of African ancestry (9.2% vs. 10.2%; *p* < 0.0001) and Americans (9.6% vs. 15.8%; *p* < 0.0001). *NPM1* mutations were detected by the DNAseq component, as these are DNA sequence mutations and not gene rearrangements, which are detected by RNAseq.

Table 1 demonstrates a comparison of the patients’ baseline characteristics.

### 3.2. Study Outcomes

A total of 633 (15.1%) of the 4206 AML cases featured NPM1 GA (NPM1mut). Short variant mutations were found in >99% of the NPM1mut AML, with the W288fs*12 frameshift base substitution accounting for 92.4% of cases. An NPM1-MLF1 fusion was identified in 1.3% of NPM1mut cases. MSI High status was not identified in any AML cases in this study (0% in both groups). HRDsig+ was also extremely uncommon in both NPM1mut and NPM1wt AML cases (0–0.1%), as was an elevated TMB (median < 1 mutation/Mb).

GA was more frequently identified in NPM1mut AML compared to the NPM1wt AML cohort, which included *DNMT3A* (39.2% vs. 12.6%; *p* < 0.0001), *FLT3* (54.5% vs. 14.7%; *p* < 0.0001), *IDH1* (16.1% vs. 5.6%; *p* < 0.0001), *IDH2* (19.0% vs. 9.0%; *p* < 0.0001), *TET2* (23.4% vs. 13.5%; *p* < 0.0001), *PTPN11* (18.3% vs. 7.5%; *p* < 0.0001), *WT1* (12.5% vs. 9.4%; *p* = 0.02), and *CEBPA* (8.2% vs. 6.4%; ns).

GA was more frequent in NPM1wt AML compared to the NPM1mut AML cohort and included *ASXL1* (17.1% vs. 3.6%; *p* < 0.0001), *BCOR* (7.5% vs. 1.6%; *p* < 0.0001), *KMT2A* (14.7% vs. 0.2%; *p* < 0.0001), *RUNX1* (22.5% vs. 1.9%; *p* < 0.0001), *STAG2* (6.9% vs. 1.6%; *p* < 0.0001), *TP53* (19.1% vs. 4.1%; *p* < 0.0001), *SRSF2* (12.3% vs. 9.8%; ns), *U2AF1* (6.8% vs. 1.3%; *p* < 0.0001), and KRAS (9.3% vs. 7.0%; *p* = 0.07).

*NRAS* and *NF1* were equally distributed in both cohorts.

Figure 2 presents the detailed genomic differences between the NPM1mut and NPM1wt cohorts. Figure 3 and Figure 4 demonstrate a significant presence of GA in the NPM1mut and NPM1wt AML cohorts, respectively.

Table 2 demonstrates a comparison of the landscape of genomic alterations in the NPM1mut and NPM1wt AML patients.

## 4. Discussion

In recent years, *NPM1* has received significant attention due to the discovery of its involvement in various human malignancies [5]. *NPM1* is now recognized as the most commonly mutated gene in patients with AML, occurring in approximately 30% of cases, and is strongly associated with de novo AML cases with a normal karyotype [45,46,47,48,49]. *NPM1* GA is relatively less common in childhood AML, ranging from 2 to 9% [50,51].

NPM1mut AML classically represents a clinically important subset of AML that is globally chemo-sensitive with a complete remission (CR) rate nearing 90%. However, despite being generally sensitive to conventional chemotherapy regimens, relapses develop in more than 50% of treated patients [36,52,53]. In AML, clusters of mutated genes are frequently observed and can significantly impact patient outcomes and drug sensitivity [54]. This is particularly relevant in NPM1mut AML and has become an area of interest with multiple studies evaluating the clinical impact of GA in NPM1mut AML [55,56,57]. In AML, the interactions between GA in *NPM1*, *KMT2A,* and menin protein have been linked to leukemogenesis and represent new potential targets for anti-tumor therapies, including menin inhibitors (such as revumenib, ziftomenib, bleximenib, and DSP-5336). The search for biomarkers to predict sensitivity to the menin inhibitors has now revolutionized our treatment approach for AML and impacted clinical outcomes, especially in the elderly population and in relapse/resistant cases [29,30,58].

Menin inhibitors are now FDA-approved in NPM1mut AML [31,59,60], *KMT2A*-rearranged, relapsed, or refractory AML [61]. Clinical trials are also evaluating the addition of menin inhibitors in combination with venetoclax and hypomethylating agents in NPM1mut and *KMT2A*-rearranged AML with preliminary high rates of CR [62], including an ongoing clinical trial (NCT05735184) investigating the safety and tolerability of ziftomenib in combination with venetoclax/azacitidine, venetoclax, or 7 + 3 in patients with AML [63].

In our study, the NPM1mut mutation rate in AML was 15.1%, which is slightly lower than what is reported in the literature (20–30%) [45,46]. This observation may reflect the fact that some of the combined DNA and RNA sequencing assay specimens were obtained after treatment or AML disease progression. In our study, NPM1mut was more frequently observed in females (53.4% vs. 41.5%; *p* < 0.0001), which is concordant with the literature [64]. Patients with NPM1mut were slightly older (62 vs. 60 y/o). *NPM1* genomic alterations were more prevalent among patients of European ancestry (77.1% vs. 68.5%; *p* < 0.0001) and slightly less common among those of African ancestry (9.2% vs. 10.2%; *p* < 0.0001) and Americans (9.6% vs. 15.8%; *p* < 0.0001). Short variant mutations represent almost all cases in *NPM1*, which were observed in >99% of the NPM1mut cohort, with the W288fs*12 frameshift base substitution accounting for 92.4% of cases, which is concordant with the literature [65]. MSI status and HRDsig—which are more relevant in solid tumors—were reported in <1% of AML cases. Testing for MSI and HRD signs in AML patients is not routinely recommended and has no implications in clinical practice.

Interestingly, in the NPM1mut cohort, genomic alterations in other genes known to be potential targets of therapy were more frequent than identified in the NPM1wt cases, such as: FLT3 (54.5% vs. 14.7%; *p* < 0.0001), IDH1 (16.1% vs. 5.6%; *p* < 0.0001), IDH2 (19.0% vs. 9.0%; *p* < 0.0001). Co-mutation patterns impact prognosis in AML, and our knowledge is evolving now that, among NPM1mut AML cases, which is generally known to be chemo-sensitive with high CR rates, other co-mutation subgroups may indicate a worse prognosis with high relapse rates, such as DNMT3A co-mutation with NPM1mut AML [66]. *DNMT3A* mutations, which encode a DNA methyltransferase, are highly recurrent in de novo AML and occur in approximately 20–25% of AML cases. *DNMT3A* mutations are independently associated with inferior survival and may further influence the prognosis of favorable mutations in AML, such as NPM1 [67,68,69,70]. *PTPN11,* which is less common in AML (approximately 5–10%), also seems to influence prognosis negatively in AML cases with concurrent *PTPN11* and *NPM1* mutations [71,72].

In our study, *DNMT3A*, which correlates with inferior outcomes, was strongly and more commonly observed with NMP1mut (39.2% vs. 12.6%; *p* < 0.0001). Additionally, *PTPN11*, which also correlates with inferior prognosis, was more commonly observed with NPM1mut AML (18.3% vs. 7.5%; *p* < 0.0001) [71]. *TET2* was more observed in NPM1mut AML (23.4% vs. 13.5%; *p* < 0.0001). *CEBPA* mutations, which are found in 10–15% of AML and generally experience a better prognosis—especially biallelic mutations—were slightly observed more with NPM1mut AML (8.2% vs. 6.4%, ns)

Chromatin spliceosome mutations were more commonly observed in NPM1wt, such as ASXL1 (17.1% vs. 3.6%; *p* < 0.0001), BCOR (7.5% vs. 1.6%; *p* < 0.0001), RUNX1 (22.5% vs. 1.9%; *p* < 0.0001), STAG2 (6.9% vs. 1.6%; *p* < 0.0001). Chromatin spliceosome mutations in AML, like ASXL1, BCOR, RUNX1, and STAG2, demonstrate inferior clinical outcomes similar to other adverse risk AML, with a high rate of relapse and poor long-term survival. These mutations are individually and collectively recognized as adverse risk group AML in the European Leukemia Society 2022 classification as well [8]. These mutations are more prevalent in older adults and secondary AML and are increasingly recognized as markers with therapeutic and prognostic implications.

*KMT2A* mutation was exclusively observed in NPM1wt (14.7% vs. 0.2%; *p* < 0.0001), suggesting that in patients with NPM1mut AML, the chances of identifying KMT2A are extremely low, at <0.5%. *TP53* mutations, which are found in 5–10% of de novo AML and up to 30–40% of therapy-related and secondary AML, represent a significant challenge in AML and MDS due to their resistance to conventional chemotherapy, including cytarabine- and anthracycline-based chemotherapy [68,72,73,74]. The rate of *TP53* mutations in our study was significantly higher in patients with NPM1wt compared to NPM1mut AML (19.1% vs. 4.1%; *p* < 0.0001), which correlates with the literature [75,76].

One main limitation of this study was that clinical outcomes like CR and OS were not available in this cohort.

## 5. Conclusions

Mutations linked to therapy targets in AML, such as *FLT3* and *IDH1/2,* are more frequently identified in NPM1mut than NPM1wt AML, whereas *KMT2A*, *TP53*, and myelodysplastic-related mutations are more frequently identified in NPM1wt AML. *DNTM3A* and *PTPN11,* which correlate with inferior outcomes, are more commonly observed in NPM1mut AML. This genomic landscape study highlights significant genomic differences between NPM1mut and NPM1wt AML patients, which may enrich our understanding of the molecular profile and mutation clusters in AML.

## Figures and Tables

**Figure 1 cancers-17-02710-f001:**
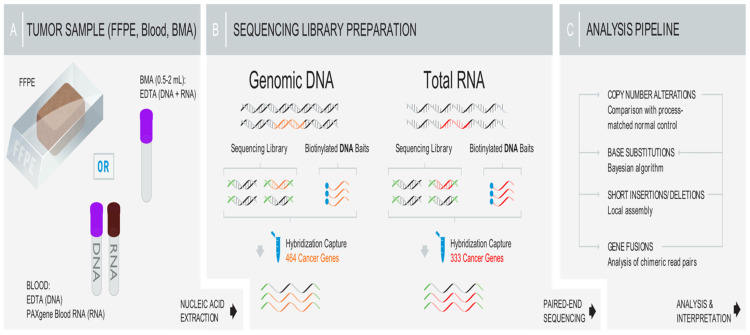
Genomic sequencing process illustration.

**Figure 2 cancers-17-02710-f002:**
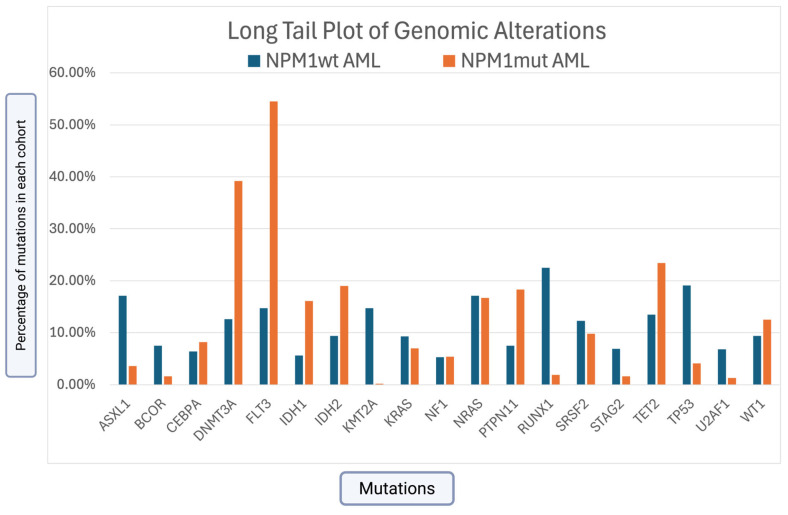
Long tail genomic alterations in NPM1mut vs. NPM1wt.

**Figure 3 cancers-17-02710-f003:**
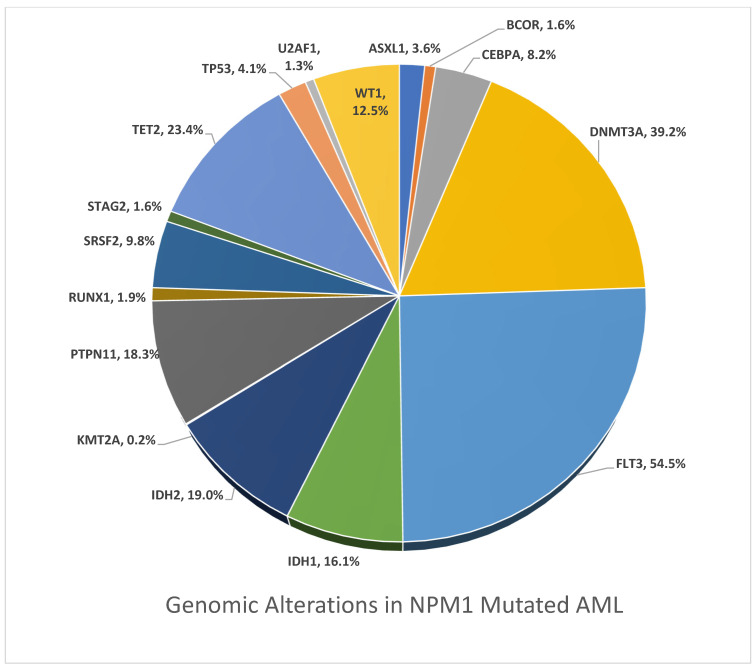
Pie chart of genomic alterations in NPM1mut AML.

**Figure 4 cancers-17-02710-f004:**
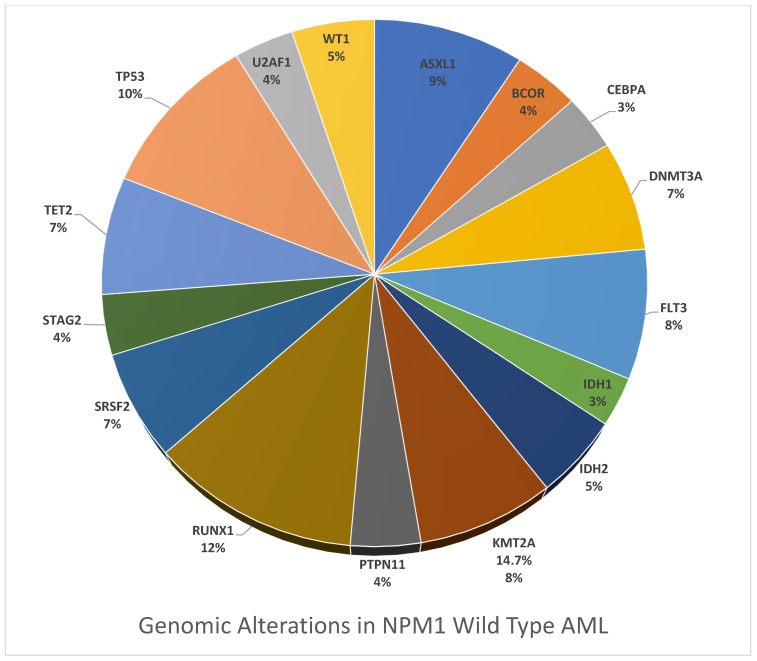
Pie chart of genomic alterations in NPM1wt AML.

**Table 1 cancers-17-02710-t001:** Baseline characteristics comparison between NPM1mut vs. NPM1wt AML patients.

Patients Characteristics	NPM1wt (*n* = 3573)	NPM1mut (*n* = 633)	*p*-Value
Sex			<0.0001
Male	2090 (58.5%)	295 (46.6%)	
Female	1483 (41.5%)	338 (53.4%)	
Age (median, range)	60 (0–89)	62 (2–89)	<0.0001
Genomic ancestry			
African	364 (10.2%)	58 (9.2%)	ns *
American	564 (15.8%)	61 (9.6%)	<0.0001
East-Asian	129 (3.6%)	14 (2.2%)	ns
European	2448 (68.5%)	488 (77.1%)	<0.0001
South-Asian	71 (2%)	12 (1.9%)	ns

* ns (not statistically significant: *p* value < 0.05).

**Table 2 cancers-17-02710-t002:** Genomic analysis comparison between NPM1mut vs. NPM1wt AML patients.

	Total Cohort (*n* = 4206)		*p*-Value ^†^
	NPM1wt (*n* = 3573)	NPM1mut (*n* = 633)	
Pathogenic genomic alterations **^††^**			
ASXL1	17.1%	3.6%	<0.0001
BCOR	7.5%	1.6%	<0.0001
CEBPA	6.4%	8.2%	ns
DNMT3A	12.6%	39.2%	<0.0001
FLT3	14.7%	54.5%	<0.0001
IDH1	5.6%	16.1%	<0.0001
IDH2	9.4%	19.0%	<0.0001
KMT2A	14.7%	0.2%	<0.0001
KRAS	9.3%	7.0%	0.07
NF1	5.3%	5.4%	ns
NRAS	17.1%	16.7%	ns
PTPN11	7.5%	18.3%	<0.0001
RUNX1	22.5%	1.9%	<0.0001
SRSF2	12.3%	9.8%	ns
STAG2	6.9%	1.6%	<0.0001
TET2	13.5%	23.4%	<0.0001
TP53	19.1%	4.1%	<0.0001
U2AF1	6.8%	1.3%	<0.0001
WT1	9.4%	12.5%	0.03
Microsatellite instability (MSI)			<0.0001
Number	3501	626	
MSI-high	0	0	1
Tumor mutational burden (TMB)			
*n*	3573	633	
Median TMB (range)	0.81	0.81	6.17 × 10^−4^
TMB ≥ 10 mut/Mb	0.3%	0.0%	ns
TMB ≥ 20 mut/Mb	0.1%	0.0%	1
Homologous recombination deficiency (HRDSIG)			
*n*	1508	188	
HRDSIG positive	0.1%	0%	1
COSMIC Trinucleotides signature			
*n*	3573	633	
Alkylating	0%	0%	ns
APOBEC	0%	0%	ns
MMR	0.7%	0.2%	ns
POLE	0%	0%	ns
Tobacco	0.1%	0%	ns
UV	0%	0%	ns

^†^ False discovery rate (FDR) corrected using Benjamini–Hochberg adjustment. ^††^ Genes only included if seen at >5% in any population. COSMIC: Catalogue of Somatic Mutations in Cancer, MMR: Mismatch repair, POLE: DNA polymerase epsilon.

## Data Availability

The raw data supporting the conclusions of this article will be made available by the authors on request.

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
