# Peer review of "A Comprehensive Genomic Analysis of Nucleophosmin (NPM1) in Acute Myeloid Leukemia"

_cancers, 2025, doi:10.3390/cancers17162710_

Round 1
Reviewer 1 Report
Comments and Suggestions for Authors
The authors were screening of mutations from 4206 AML patients to “conduct a comprehensive landscape genomic analysis”. The topic the study is interesting and up-to-date. The large patient cohort enable reliable conclusions of how the mutations affect the AML carcinogenesis and therapy.
The results section is narrow and does not offer adequate amount of information. The study would have been significantly more interesting if the authors could had compared the outcome of the patients’.
Author Response
"The authors were screening of mutations from 4206 AML patients to “conduct a comprehensive landscape genomic analysis”. The topic the study is interesting and up-to-date. The large patient cohort enable reliable conclusions of how the mutations affect the AML carcinogenesis and therapy.
The results section is narrow and does not offer adequate amount of information. The study would have been significantly more interesting if the authors could had compared the outcome of the patients’. "
Authors response: We totally appreciate the comment, however, clinical outcomes are limited from the foundation medicine database that was used in our study. Results section was expanded and discussion sections was revised thoroughly and updated to highlight clinical outcomes of different mutation clusters that was observed in our study.
Reviewer 2 Report
Comments and Suggestions for Authors
The manuscript titled "Characterizing Nucleophosmin (NPM1) Genomic Alterations in Acute Myeloid Leukemia: A Comprehensive Genomic Study" by Batayneh et al. represents an extensive genomic profiling comparison between NPM1mut and NPM1wt AML cases. While the study is significant and presents clinically valuable data, better clarity of the assay and the scientific rigor, enhanced clarity with better data presentation would be beneficial for the manuscript. Below I have discussed pointwise the areas of the manuscript that can be improved:
(1) The methodology part can be benefitted from additional details about FoundationOne heme assay like how RNA vs DNA data were integrated in variant calling, the details about assay reproducibility, sequencing depth, sample quality control etc.
(2) The authors are requested to represent the figures in a better way for improved clarification. Figure 2 lacks a clear Y-axis legend making it difficult for interpretation. Table 1 and 2 contain dense information but can be arranged into heatmaps or bar graphs for better understanding.
(3) More details should be provided on how Benjamini-Hochberg correction was applied and which comparisons it covered. For mutation frequency differences, please include confidence intervals.
(4) The manuscript would benefit from some grammatical corrections. Some phrases are redundant and repeated. Sometimes the sentences are used in an informal way like “'was seen more in females' should be 'was more frequently observed in females'; 'seen in >99%' is informal use 'observed in over 99%'. The discussion contains several run-on sentences that should be split for clarity.
Addressing the upper-mentioned concerns would make the manuscript suitable for publication in the journal.
Comments on the Quality of English Language
The manuscript would benefit from some grammatical corrections. Some phrases are redundant and repeated. Sometimes the sentences are used in an informal way like “'was seen more in females' should be 'was more frequently observed in females'; 'seen in >99%' is informal use 'observed in over 99%'. The discussion contains several run-on sentences that should be split for clarity.
Author Response
(1) The methodology part can be benefitted from additional details about FoundationOne heme assay like how RNA vs DNA data were integrated in variant calling, the details about assay reproducibility, sequencing depth, sample quality control etc.
Authors response: this was addressed as suggested in the methods sections.
(2) The authors are requested to represent the figures in a better way for improved clarification. Figure 2 lacks a clear Y-axis legend making it difficult for interpretation. Table 1 and 2 contain dense information but can be arranged into heatmaps or bar graphs for better understanding.
authors response: Figures added to better represent data. X and Y axis legend added to Figure 2. Bar graph is available as a representation of the data in figure 2.
(3) More details should be provided on how Benjamini-Hochberg correction was applied and which comparisons it covered. For mutation frequency differences, please include confidence intervals.
Authors response:
Clarification added in the methods section explaining how the Benjamini-Hochberg procedure was used. During the limited time to revise the manuscript, we were unable to add CI, however if this is absolutely needed we would be happy to add this if deadline can be extended.
(4) The manuscript would benefit from some grammatical corrections. Some phrases are redundant and repeated. Sometimes the sentences are used in an informal way like “'was seen more in females' should be 'was more frequently observed in females'; 'seen in >99%' is informal use 'observed in over 99%'. The discussion contains several run-on sentences that should be split for clarity.
Authors response:
We totally appreciate this comment. Manuscript was revised thoroughly to address this matter.
Reviewer 3 Report
Comments and Suggestions for Authors
The main of the study was a comprehensive landscape genomic analysis of
key genomic alterations in AML.
Hence, the title is not appropriate: title should be altered to reflect the main aim. It can be re-titled as 'A comprehensive genomic analysis in AML'.
The association to NPM1 mut versus NPM1wt with other genomic alterations is a secondary analysis and does not appear to be the primary goal.
Comments on the Quality of English LanguageSeveral incomplete phrases and sentences need the attention of the authors - the meaning convey is not clear.
It should be thoroughly revised to make sense to an English language reader. Please avoid using paragraph length single sentences. Please break them into short sentences, each conveying a concept.
Author Response
Comment 1:
"The main of the study was a comprehensive landscape genomic analysis of
key genomic alterations in AML.
Hence, the title is not appropriate: title should be altered to reflect the main aim. It can be re-titled as 'A comprehensive genomic analysis in AML'.
The association to NPM1 mut versus NPM1wt with other genomic alterations is a secondary analysis and does not appear to be the primary goal."
Authors response: We appreciate the suggestion. Title is edited as suggested.
Comment 2 :
"Several incomplete phrases and sentences need the attention of the authors - the meaning convey is not clear.
It should be thoroughly revised to make sense to an English language reader. Please avoid using paragraph length single sentences. Please break them into short sentences, each conveying a concept."
Authors response: Addressed thoroughly in the revised manuscript
Round 2
Reviewer 1 Report
Comments and Suggestions for Authors
N/A
Reviewer 3 Report
Comments and Suggestions for Authors
Thank you for the replies and the corrections